# Symptom-Level Disability Status Assessed with an Electronic Unsupervised Patient-Reported Expanded Disability Status Scale (ePR-EDSS) in Multiple Sclerosis Patients—The Example of Croatia

**DOI:** 10.3390/jcm11144081

**Published:** 2022-07-14

**Authors:** Ana Jerković, Sanda Pavelin, Joško Šoda, Igor Vujović, Maja Rogić Vidaković

**Affiliations:** 1Laboratory for Human and Experimental Neurophysiology (LAHEN), Department of Neuroscience, School of Medicine, University of Split, 21000 Split, Croatia; anasuto@gmail.com; 2Department of Neurology, University Hospital of Split, 21000 Split, Croatia; spavelin@gmail.com; 3Department of Marine Electrical Engineering and Information Technologies, Faculty of Maritime Studies, University of Split, 21000 Split, Croatia; jsoda@pfst.hr (J.Š.); ivujovic@pfst.hr (I.V.)

**Keywords:** MS, multiple sclerosis, relapsing-remitting, primary progressive multiple sclerosis, secondary progressive multiple sclerosis, expanded disability status scale, EDSS, ePR-EDSS, eHealth

## Abstract

The present study aimed to apply an electronic, unsupervised patient-reported Expanded Disability Status Scale (ePR-EDSS) to investigate disability severity in people with multiple sclerosis (pwMS) as a case study in Croatia in 2021, including demographic and comorbidity characteristics and multiple sclerosis (MS) disease-related factors. The cross-sectional study was conducted as an online survey from 4 October 2021 to 31 December 2021. Symptom-level disability status was assessed with ePR-EDSS for MS capturing MS-related disability across the spectrum of severity.The study enrolled 147 pwMS patients, of which 84% were women. The mean age ± standard deviation in the sample was 41.1 ± 11.3, and the mean disease duration was 8.5 ± 7.4 years, with a median EDSS score of 3.0 (range, 0–8). The distribution of the participants according to clinical forms of MS was as follows: 71% had relapsing-remitting MS, 13% had primary progressive MS, 4% had secondary progressive PMS, and 12% did not provide information on their MS type. Twenty-nine point two percent (29.2%) of the participants had comorbidities in addition to MS. EDSS scores indicate significant differences with regard to age (t = −3.51, *p* < 0.001), gender (χ^2^ = 8.04, *p* < 0.01), and immunomodulatory drug use (χ^2^ = 5.89, *p* < 0.05). An ePR-EDSS analysis of disability symptoms showed a significant difference in symptoms with regard to strength, sensation, coordination, vision, fatigue, mobility, and overall wellness among MS types. Participants with PPMS and SPMS were older on average, had higher EDSS, and had more pronounced symptoms of disability measured with ePR-EDSS compared to those with RRMS. Application of ePR-EDSS shows it to be a reliable eHealth tool for clinical assessment of pwMS disability status, and future studies should correlate it with standard self-report scales capturing MS symptoms such as fatigue, depression, anxiety, and stress.

## 1. Introduction

Multiple sclerosis (MS) is a potentially disabling disease of the brain and spinal cord, and it is twice as common in women as in men. The disease incidence is about 7 per 100,000 inhabitants each year, and the prevalence in European countries ranges between 80 and 120 per 100,000 inhabitants [1]. The total number of people with MS (pwMS) in Croatia is 6160, and the prevalence is between 144 per 100,000 inhabitants [2,3]. MS is considered an autoimmune disease, and the main agents responsible for its development include exogenous, environmental, and genetic factors [4]. Gender differences include earlier disease onset and more frequent relapses in women. Significant influence, both on disease prevalence and on outcomes, can be due to hormone-related physiological conditions in women, such as puberty, pregnancy, and menopause [5]. Even though the pathogenesis of MS requires both genetic predisposition and specific environmental exposure, genetic susceptibility is rare in the MS population (<7.3%) and requires specific combinations of non-additive genetic risk factors. Moreover, sufficient environmental exposure is expected in at least 76% of susceptible individuals [6].

The relapsing-remitting form of the disease (RRMS) is the most common form, with a prevalence of about 80 to 90% of pwMS. The primary-progressive form of the disease (PPMS) is significantly less common and occurs in 10% of patients, while further disease progression indicates the transition from the relapsing-remitting to the secondary-progressive form of the disease (SPMS) [7]. Diagnosis of MS is based on the recommendations of the revised McDonald criteria valid from 2018 [8,9,10]. Further, the Expanded Disability Status Scale (EDSS) is a frequently used measure to quantify neurological disability due to MS and evaluate disability changes over time from the physician’s point of view [11,12,13]. EDSS scale requires an in-person assessment and suffers from having high inter- and intra-rater variability, especially at the lower disability levels [14,15]. In addition, in population-based surveys such as cross-sectional studies it is not feasible to perform in-person evaluations of disability status in pwMS. Hohol et al. [16] developed the Patient Determined Disease Steps (PDDS) scale for assessing disability in pwMS when direct clinical evaluation is not possible. The Italian version of PDDS confirms the equivalence of the PDDS with EDSS [17]. Similarly, an electronically administered patient-reported EDSS (ePR-EDSS) [18] was recently validated across a wide EDSS range, permitting assessment of disability status during clinical trials, increasing patients’ access to MS clinical research, and potentially decreasing utilization of resources and personnel time. Psychometric properties of ePR-EDSS were determined by splitting the participants into two validation cohorts evenly distributed in the EDSS score. Each participant completed an ePR-EDSS [18], and clinician completed a neurostatus EDSS evaluation [15]. The ePR-EDSS correlates very well with standard EDSS (Neurostatus-EDSS) for disability assessment for pwMS, with good agreement even at lower EDSS levels.

The present study aimed to apply ePR-EDSS to investigate disability severity in pwMS patients in 2021 taking Croatia as an example, including demographic and comorbidity characteristics and MS disease-related factors.

## 2. Materials and Methods

### 2.1. General Procedures

The study was a cross-sectional survey conducted via an online platform from 4 October 2021 to 31 December 2021. PwMS patients were recruited by advertising through the Association of Multiple Sclerosis Societies of Croatia (AMSSC) and MS TIM Croatia (a non-governmental association for people with pwMS).

### 2.2. Participants

The 147 participants filled out self-administered questionnaires. Duplicated answers and data of pwMS patients who were not residents of Croatia were eliminated from the study. All the procedures performed in the studies that involved human participants followed the ethical standards of the Ethics Committee of the University of Split, School of Medicine (Class: 003-08/21-03/0003; No.: 2181-198-03-04-21-0039, approved on 10 March 2020 and 23 April 2021 -annex), the Ethical Committee of the University Hospital of Split (Class: 500-03/20-01/06, No.: 2181-147-01/06/M.S.-20-2, approved on 27 January 2020), and the 1964 Helsinki Declaration and its later amendments or comparable ethical standards.

### 2.3. Methods

The data collection included the following information: demographic characteristics (age, gender, height, hand dominance, body mass index (BMI), Croatian county of residence, householding information, education, marriage status, work status, family income), and MS disease-related factors (MS type, assistive technology, EDSS score, information on corticosteroid and immunomodulatory treatment and other recommended medications for MS, duration of immunomodulation therapy, and comorbidity and medication prescribed). The data included disability status self-assessed with an electronic, unsupervised ePR-EDSS for MS capturing MS-related disability across the spectrum of severity [18]. The ePR-EDSS consists of 23 questions about pwMS, including the following topics: (1) overall wellness—functional abilities; (2) mobility—the ability to walk; (3) sensation; (4) strength of the arms and legs; (5) coordination; (6) vision; (7) muscle weakness in the face, sensation in the face, ability to speak, ability to swallow liquids and solids, and hearing problems; (8) bowel and bladder function; and (9) mood changes and thinking ability (cognition, fatigue, depression). It took 7–12 min to complete the ePR-EDSS (15) and approximately five minutes to complete demographic, comorbidity, and MS disease-related information. The participants answered each question on a three-, four-, five-, or six-point (0–3; 0–4; 0–5; 0–6) Likert scale; they also answered questions with regard to disability. The overall ePR-EDSS score was not calculated in the current study, only the symptom-level analysis of the disability status of pwMS. A high correlation of ePR-EDSS with the EDSS (Neurostatus- EDSS) (standard assessment tool) score was reported previously [15,18].

### 2.4. Statystical Analysis

Univariate analyses were performed using the Chi-square test (χ^2^) and Fisher’s exact test (*p*) for categorical variables. The one-way ANOVA test (F value), the Kruskal–Wallis test (H value), and Student’s *t*-test were used for continuous and ordinal scale variables. Levene’s test was used to assess the assumption of the equality of variances between groups. When calculating differences between relevant demographic and disease-related parameters, EDSS scores are divided into two categories (0–3.5—depending on functional system; 4–8—dependence on help) [19]. Furthermore, when calculating differences between relevant disease–related parameters, SPMS and PPMS types are considered as one category due to the small sample of these two MS types. Descriptive statistics of relevant participants’ characteristics and applied scales were summarized by N, percentage, mean and standard deviations, and median and interquartile range (IQR). A threshold of *p* < 0.05 was used for determining the level of effect significance. Data analysis was performed using the software Statistica 12 (TIBCO Software Inc., Palo Alto, CA, USA).

## 3. Results

### 3.1. Demographic Characteristics, Comorbidity, and MS Disease-Related Factors

In total, 147 pwMS patients filled out questionnaires. Eighty-four percent (84%) of the patients were women, with a mean age of 41.1 ± 11.3 years, and 16% were men, with a mean age of 45.3 ± 10.4 years. The demographic and disease-related factors for pwMS are presented in Table 1. Most of the patients lived in Zagreb (23.1%) or Split-Dalmatia counties (27.2%). More than half (54.4%) lived in apartments, 45.6% lived in houses, and a majority (69.4%) lived with a spouse. Seventy-seven percent (77%) lived in marriage or cohabitation. Fifty-two percent (52%) finished high school, 32% a university degree, 7.5% professional study, 4.1% specialist professional study, 3.4% postgraduate specialist study, and 0.7% doctoral study. The majority (50.3%) were employed, 18.4% were pensioners, and 11.5% were temporiarily on sick leave.

Seventy-one percent (71%) of the study participants had RRMS, 13% had PPMS, 4% had SPMS, and 12% did not provide their MS type (*MS type not known*). On average, SPMS patients had the longest disease duration (14.8 years; Table 1).

Nineteen percent (19%) of the participants did not know their score on the EDSS scale. The average result on the EDSS scale for all the participants was 3.0 (IQR = 3.0), and pwMS with PPMS and SPMS, in general, had higher EDSS in regard to RRMS and pwMS with *MS type not known*. Among participants who underwent assistive technology, 61% needed glasses and 20% needed crutches. Twenty-eight percent (28%) of the participants did not use any aid for their everyday activities. Over 90% received corticosteroid treatment, and of these, 45.6% received it more than three times. Sixty-eight percent (68%) received immunomodulation therapy. Participants with RRMS received immunomodulatory therapy in 70.5% of cases, those with PPMS in 78.9% of cases, those with *MS type not known* in 58.8% of cases, and those with SPMS in 16.7% of cases. The most commonly taken immunomodulatory drug was Ocrelizumab (Ocrevus) (20%), followed by Glatiramer-Acetate (15.5%) and Fingolimod (15.5%) (Table 2). Participants with RRMS took immunomodulation therapy mostly for one to three years (38%) or one to ten years (38.8%). One participant with SPMS took the therapy for one to seven days, and 63% of PPMS for one to three years. For *MS type not known*, the duration of therapy was different for each participant: 11% took 1–6 months of therapy, 22% took 6–12 months, 11% took 1–3 years, 22% took 3–5 years, and 11% took 5–10 years. Among those participants not receiving immunomodulation, for 85% the physician recommended supplements such as vitamin D or B (52%) or magnesium (41%).

From a total of 147 participants, 29.2% had comorbidities, of which the most common were endocrine, nutritional, and metabolic diseases (44.1%) and diseases of the circulatory system (13.9%) (Table 3). With regard to age, comorbidities occurred more frequently in the early forties (>40 years); this was true for 70% of the participants. Among participants 30 to 39 years of age, 19.5% had comorbidity, and among those 20–29 years of age, about 9% had it.

### 3.2. MS Type and EDSS-Related Differences

A significant difference was found in the age of the participants with regard to their MS type (Table 1), where participants with RRMS were younger than those with SPMS or PPMS (F = 6.12; *p* < 0.01). A significant difference was also found in the EDSS score (χ^2^ = 18.5, *p* < 0.001) (Table 1), where participants with SPMS and PPMS had higher EDSS scores than did participants with RRMS and *MS type not known*. Regarding disability status (EDSS) range (Table 4), significantly higher EDSS (4–8) was found in older participants (t = −3.5, *p* < 0.001). Moreover, significantly higher EDSS was found in male participants (χ^2^ = 8.04; *p* < 0.01), indicating that more male participants had higher disability status compared to those with lower EDSS (0–3.5), and participants with higher EDSS (4–8) received significantly fewer immunomodulation drugs χ^2^ = 5.89; *p* < 0.05) (Table 4).

### 3.3. ePR-EDSS Disability Status and Symptom-Level Analysis

Concerning ePR-EDSS disability status, the symptom-level analysis showed significant differences in overall wellness-functional abilities with regard to MS types (H = 18.6; *p* < 0.001) (Table 5). The results indicated lower overall wellness in participants with SPMS and PPMS compared to those with RRMS and *MS type not known* (Table 5; Figure 1). The results confirmed that mobility (ability to walk) becomes lower as MS progresses (H_ability to walk_ = 21.3, *p* < 0.001; H_no use an aid_ = 9.17, *p* < 0.001; H_without aid_ = 918.02, *p* < 0.01; H_use a cane_… = 17.36, *p* < 0.001; H_use a walker_… = 12.38, *p* < 0.001; H_use a wheelchair_… = 7.12, *p* < 0.001) (Table 5. Furthermore, significantly more pronounced severity differences were found in PPMS and SPMS compared to RRMS and *MS type not known* in other categories of disability symptoms (Table 5), such as sensation in the lower extremities (H_right foot or leg_ = 7.13, *p* < 0.05; H_left foot or leg_ = 6.07, *p* < 0.05); strength in the lower extremities to rise (H_right leg_ = 16.93, *p* < 0.01; H_left leg_ = 12.16, *p* < *0*.01); strength to raise the right arm (H = 13.64, *p* < 0.01); muscle stiffness in the lower extremities (H_right leg_ = 14.69, *p* < 0.01; H_left leg_ = 9.51, *p* < 0.01); coordination (legs—H = 14.36, *p* < 0.01; balance with standing—H = 13.85, *p* < 0.01; balance with walking—H = 21.97, *p* < 0.001); vision problems in the right (H = 8.46, *p* < 0.01) and left eye (H = 7.21, *p* < 0.05); muscle weakness in both sides of the face (right—H = 11.18, *p* < 0.01 and left—H = 7.73, *p* < 0.05); feeling problems in the right side of the face (H = 6.9, *p* < 0.05); urinary urgency (H = 12.69, *p* < 0.01); and fatigue (H = 9.37, *p* < 0.01) (Table 5; Figure 2).

Figure 1 presents average percentages of overall wellness-functional abilities. About 25% of participants with SPMS and PPMS were severely limited compared to 6% of participants with RRMS. Figure 2 indicates fatigue level with regard to MS type. Two degrees of fatigue (fatigue that affects less than half of daily activities and severe fatigue that affects more than half of daily activities) were expressed in over 50% of people with PPMS and SPMS.

## 4. Discussion

The present cross-sectional study provides results on the degree of disability of pwMS and insight into demographic and clinical data related to MS in a Croatian sample of pwMS patients in 2021. In the present study sample, 84% of the participants were women, and there were no differences in the gender ratio of pwMS with regard to MS type. The study results agree with previous findings on the higher prevalence ratio of women to men with regard to MS (2.3–3.5:1) [20]. Women generally have an earlier onset of the disease, with a slightly lower prevalence of PPMS course, and show, in general, less progression in the disability than men [5,20,21,22]. The male participants with RRMS were mostly diagnosed in their early twenties, and for PPMS the prevalence for males was greater from the forties onward. In general, the transition to a progressive course of the disease seems to be age-dependent [23]. Participants with SPMS and PPMS were older and had higher EDDS scores than those with RRMS. This is in line with previous studies of MS progression, which found that the age of transition occurs around the forties [24,25,26] and that other chronic diseases also play a role in the disease course. In the present study, comorbidities that occur in pwMS were mostly endocrine, nutritional, and metabolic diseases (e.g., Hashimoto, hypothyroidism, hyperthyroidism, hyperlipidemia, obesity, diabetes, osteoporosis, osteopenia, Gilbert syndrome, and hypovitaminosis) and diseases of the circulatory system (e.g., Raynaud’s phenomenon, essential hypertension, deep vein thrombosis, angina pectoris, tachycardia, atrial septal aneurysm, arrhythmia, and atrial fibrillation). An increase in cardiovascular comorbidities could be associated with the level of disability, reflecting reduced physical activity and more sedentary behavior, causing general health to worsen and increasing healthcare resources consumption [26,27,28]. Moreover, cardiovascular dysfunction in pwMS often co-occurs with comorbid metabolic conditions (e.g., diabetes, hypertension, and hyperlipidemia) [25,29].

The EDSS is not the most sensitive scale for assessing the progression of MS [30], nor is it always sensitive enough to detect differences in clinical forms of MS [19]. The ePR-EDSS is a new electronic patient-reported Expanded Disability Status Scale that is highly correlated with the standard EDSS scale [18] and provides a detailed analysis of common pwMS symptoms and symptoms that may be under-recognized and therefore not observed directly [19]. In the present study, symptoms of disability were analyzed using ePR-EDSS to identify the disability status of pwMS from the patient’s perspective. One of the symptoms is overall wellness, which was observed to be lower in participants with SPMS and PPMS than in those with RRMS. Overall wellness is defined as the ability to carry out usual daily activities without limitation. Because of the illness progression, SPMS and PPMS tend to affect the patient’s overall ability to function more than RRMS does. The symptom that affects key domains of well-being in progressive forms of MS is fatigue [31]. Watson et al. [31] showed that nearly all patients reported that their fatigue made it difficult to complete activities of daily living. Fatigue is a disabling symptom defined as an overwhelming sense of tiredness, exhaustion, or lack of energy, and up to 40% of pwMS patients identify it as their most disabling symptom [32]. The findings in the present study are in accord with previous results indicating higher fatigue levels in progressive forms of MS (PPMS and SPMS). Furthermore, mobility is the most frequently affected function in MS, and the results confirmed that mobility is lower with the progressive types of MS. Concerning other symptoms, the study findings indicate that sensation in the lower extremities, strength in the lower extremities and in the right arm to rise, muscle stiffness in the lower extremities, and coordination and balance when walking are lower in SPMS and PPMS. PwMS patients are generally physically less active, and this way of life results in a decrease in functional activities, loss of mobility, and balance problems [33]. Other symptoms that occur more frequently in SPMS and PPMS can also trigger higher fatigue. Lower overall wellness can include such symptoms as vision problems in both eyes, muscle weakness on both sides of the face, and urinary urgency; all these symptoms were found to be more severe in SPMS and PPMS.

The present study has limitations that need to be considered, such as a smaller number of pwMS patients with different MS types. Future studies should include the overall ePR-EDSS score and correlate ePR-EDSS scores with clinically used EDSS scores. The second limitation of the study is that it was not possible to request an official report of the status of MS (including official EDSS score) by the neurologist examination because the study was conducted online. We believe that this shortcoming did not affect the data since the participants were members of official associations for people with pwMS. Future studies could set up longitudinal studies to investigate the efficiency of ePR-EDSS in monitoring disability progression in pwMS over time. We suggest investigating disability status in pwMS so as to have objective instruments for the assessment of fatigue (i.e., Fatigue Severity Scale, FSS) and walking scales (Multiple Sclerosis Walking Scale, MSWS-12), thus enabling to correlate ePR-EDSS with subjective self-report scales [34,35,36].

As a convenient tool to longitudinally monitor changes in MS-related disability, ePR EDSS can aid in digital therapeutics and self-care management. The use of information and communication technology can help pwMS patients manage their disability symptoms [37,38]. Likewise, use by patients of self-rated validated scales for monitoring MS-related symptoms such as walking, fatigue, or the impact of MS on physical and psychological functioning [34,37,38,39,40] can perform a similar function.

## 5. Conclusions

This study applied a recently validated electronically administered patient-reported EDSS (ePR-EDSS) questionnaire [18] that permitted assessment of disability status and health condition (eHealth) of pwMS patients in Croatia in 2021, including demographic and comorbidity characteristics and MS disease-related factors. The questionnaire showed more pronounced problems in patients with PPMS and SPMS than in those with RRMS as regards overall wellness-functional abilities, mobility, sensation and strength in the legs, strength in the right arm, coordination, vision, muscle weakness in both sides of the face, feeling in the right side of the face, urinary urgency, and fatigue.

## Figures and Tables

**Figure 1 jcm-11-04081-f001:**
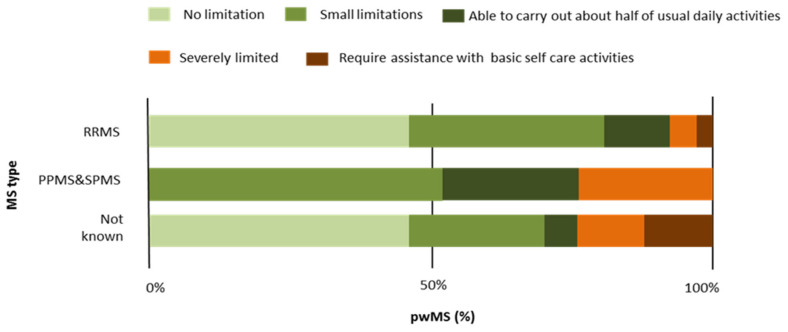
Average percentages of overall wellness (functional abilities) with regard to MS type.

**Figure 2 jcm-11-04081-f002:**
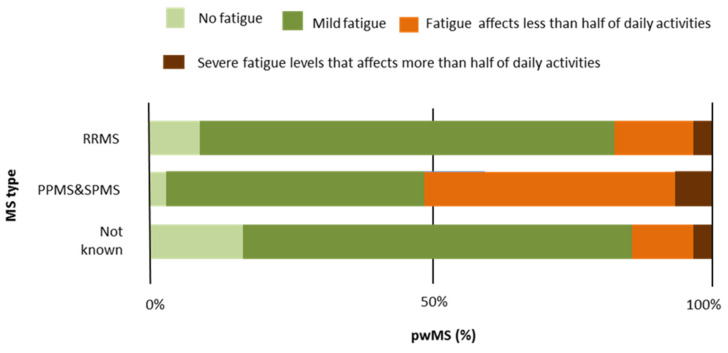
Average percentages of fatigue with regard to MS type.

**Table 1 jcm-11-04081-t001:** Differences in disease-related characteristics, considering MS type.

	RRMS	PPMS	SPMS	*MS Type Not Known*	Total	Test	*p*
N	105	19	6	17	147	-	-
Age in years (mean ± SD)	41.1 ± 1.9	47.1 ± 7.7	49.6 ± 9.1	42.9 ± 9.3	41.1 ± 11.3	F = 6.12	0.009 **
Female/Male (%)	80/20	84/16	100/0	95/5	84/16	χ^2^ = 2.45	0.27
Hand dominance-right hand (%)	92.4	94.7	100	82.4	91.8	-	-
BMI (mean ± SD)	24.5 ± 4.2	24.6 ± 5.9	24.3 ± 4.8	21.3 ± 2.9	19.6 ± 10.4	-	-
Years of MS disease (mean ± SD)	7.1 ± 1.7	5.2 ± 5.5	14.8 ± 7.4	7.2 ± 8.4	8.5 ± 7.4	F = 0.29	0.48
EDSS (median(IQR), range 0–8	3(2.5)	5.5(3)	4.7(1.5)	1.5(4.5)	3.0(3.0)	χ^2^ = 18.5	<0.001 **
Corticosteroid treatment (%)	90.4	89.5	100	94.1	91.2	χ^2^ = 0.26	0.87
Immunomodulatory therapy (%)	70.5	78.9	16.7	58.8	68.0	χ^2^ = 1.73	0.56

Abbreviations: RRMS, relapsing-remitting multiple sclerosis; PPMS, primary progressive multiple sclerosis; SPMS, secondary progressive multiple sclerosis; *MS type not known*, MS type not provided by the participant; BMI, body mass index; EDSS, Expanded Disability Status Scale; **—*p* < 0.01. Note: In calculating differences between MS types, SPMS and PPMS are considered one category due to the smaller sample.

**Table 2 jcm-11-04081-t002:** Percentages of different immunomodulation drugs used by the participants (N = 85).

Immunomodulation Medication	RRMS *n* = 64 *n*/%	PPMS *n* = 11 *n*/%	SPMS *n* = 1 *n*/%	*MS Type Not Known*, *n* = 9 *n*/%	Total *n* = 85 *n*/%
Ocrelizumab (Ocrevus)	14/21.8	3/27.2	-	1/11.1	17/20.0
Glatiramer-Acetate (Copaxone, Glatopa, Remurel)	7/10.9	3/27.2	-	2/22.2	13/15.5
Fingolimod (Gilenya)	8/12.5	1/9.1	-	2/22.2	13/15.5
Dimethyl Fumarate (Tecfidera)	9/14.2	-	1/100	1/11.1	10/11.6
Interferon (ifn beta-1a) (Rebif)	7/10.9	-	-	1/11.1	10/11.6
Peginterferon (beta-1a) (Plegridy)	6/9.3	1/9.1	-	-	6/7.8
Teriflunomid (Aubagio)	2/3.1	2/18.2	-	-	4/4.6
Interferon (ifn beta-1b) (Betaferon)	3/4.7	-	-	1/11.1	4/4.6
Natalizumab (Tysabri)	3/4.7	-	-	-	3/3.3
Cladribine (Leustatin, Mavenclad)	2/3.1	-	-	-	3/3.3
Alemtuzumab (Campath, Lemtrada)	-	1/9.1		1/11.1	2/2.3

**Table 3 jcm-11-04081-t003:** Percentages of comorbidities in the participants (N = 43).

Other Chronic Diseases (Not MS)	RRMS *n* = 35 *n*/%	PPMS *n* = 5 *n*/%	SPMS *n* = 2 *n*/%	*MS Type Not Known*, *n* = 1 *n*/%	Total *n* = 43 *n*/%
Endocrine, nutritional and metabolic diseases	15/42.8	4/80.0	-	-	19/44.1
Diseases of the circulatory system	5/14.3	1/20.0	-	-	6/13.9
Diseases of the nervous system	4/11.4	1/20.0	1/50.0	-	5/11.6
Diseases of the digestive system	4/11.4	-	-	-	4/9.3
Diseases of the respiratory system	1/2.8	-	1/50.0	1/100	3/6.9
Musculoskelet disorders	2/5.7	-	1/50.0	-	3/6.9
Diseases of the eye	3/8.5	-	-	-	3/6.9
Mental and behavioural disorders	2/5.7	-	1/50.0	-	3/6.9
Neoplasms	2/5.7	-	-	-	2/4.6
Diseases of the genitourinary system	1/2.8	-	-	-	1/2.3
Diseases of the skin	1/2.8	-	-	-	1/2.3

Abbreviations: The results are expressed in percentages. Some of the participants had several comorbidities, but the percentages are calculated in regard to chronic disease. For example, for SPMS, one subject had diseases of the respiratory system and musculoskeletal disorders combined. RRMS, relapsing-remitting multiple sclerosis; PPMS, primary progressive multiple sclerosis; SPMS, secondary-progressive multiple sclerosis; *MS type not known* = MS type not provided by the subject.

**Table 4 jcm-11-04081-t004:** Differences regarding disability status (EDSS) range.

EDSS Range	EDSS 0–3.5	EDSS 4–8	Test	*p*
N (%)	69 (58.4)	49 (41.6)	-	-
Age (mean ± SD)	39.7 ± 8.5	46.0 ± 12.5	t = −3.51	<0.001 **
Male/female (%)	8.7/91.3	28.6/71.4	χ^2^ = 8.04	0.01 **
Years of MS disease (mean ± SD)	8.8 ± 7.6	10.4 ± 7.2	t = −1.71	0.33
Corticosteroid treatment (y/*n*, %)	91.3/8.7	97.9/2.1	χ^2^ = 2.27	0.13
Immunomodulatory drug (y/*n*, %)	79.7/20.3	59.1/40.9	χ^2^ = 5.89	0.02 *

Abbreviations: EDSS, Expanded Disability Status Scale; *—*p* < 0.05; **—*p* < 0.01.

**Table 5 jcm-11-04081-t005:** Disability status and symptom-level analysis with regard to MS type.

	RRMS *n* = 105	PPMS and SPMS *n* = 25	*Ms Type Not Know n* = 17	Total *n* = 147	Test (H Value)	*p*
Overall wellness—functional abilities (1–5) *	2.0 (1.0)	2.0 (2.0)	2.0 (2.0)	2.0 (2.0)	18.6	<0.001
Mobility-ability to walk						
Ability to walk (1–4)	1.0 (1.0)	2.0 (0)	1.0 1.(0)	2.0 1.(0)	21.3	<0.001
Ability to walk without aid (1–4)	2.0 (3.0)	5.0 (2.0)	5.0 (0)	3.0 (4.0)	9.17	0.01 **
Ability to walk without aid (% of time)	1.0 (0.25)	0.2 (0.75)	1.0 (0.7)	1.0 (0.5)	18.02	<0.001
Use a cane, a single crutch, or hold onto another person (% of time)	0 (0.3)	0.8 (0.7)	0 (1.0)	0.1(0.8)	17.36	<0.001
Use a walker or other bilateral support (% of time)	0 (0)	0.5 (1.0)	0 (0.9)	0 (0.3)	12.38	<0.001
Use a wheelchair (% of time)	0 (0)	0 (0.3)	0 (0)	0 (0)	7.12	<0.02 *
Sensation						
Right hand or arm (1–6)	2.0 (1.0)	2.0 (2.0)	2.0 (2.0)	2.0 (1.0)	6.04	0.05
Left hand or arm (1–6)	1.0 (1.0)	2.0 (1.0)	2.0 (1.0)	2.0 (1.0)	2.72	0.25
Right foot or leg (1–6)	2.0 (1.0)	2.0 (1.0)	2.0 (1.0)	2.0 (2.0)	7.13	0.028 *
Left foot or leg (1–6)	2.0 (1.0)	2.0 (1.0)	2.0 (1.0)	2.0 (1.0)	6.07	0.04 *
Strength ^†^						
Right arm (1–6)	1.0 (1.0)	2.0 (1.0)	2.0 (2.0)	1.0 (1.0)	13.64	0.01 **
Left arm (1–6)	1.0 (1.0)	1.0 (1.0)	1.0 (1.0)	1.0 (1.0)	2.44	0.29
Right leg (1–6)	2.0 (1.0)	2.0 (2.0)	2.0 (2.0)	2.0 (1.0)	16.93	<0.001
Left leg (1–6)	2.0 (1.0)	2.0 (1.0)	2.0 (1.0)	2.0 (1.0)	12.16	<0.01 **
Strength ^††^						
Right arm (1–5)	1.0 (1.0)	2.0 (1.0)	2.0 (1.0)	1.0 (1.0)	2.78	0.24
Left arm (1–5)	1.0 (1.0)	1.0 (1.0)	1.0 (1.0)	1.0 (1.0)	0.65	0.71
Right leg (1–5)	2.0 (1.0)	2.0 (1.0)	2.0 (1.0)	2.0 (2.0)	14.96	<0.001
Left leg (1–5)	2.0 (1.0)	2.0 (1.0)	2.0 (1.0)	2.0 (1.0)	9.51	0.01 **
Coordination						
Arms (1–5)	2.0 (2.0)	2.0 (1.0)	2.0 (1.0)	2.0 (2.0)	4.08	0.13
Legs (1–5)	2.0 (2.0)	3.0 (2.0)	3.0 (2.0)	2.0 (2.0)	14.36	<0.001
Balance when standing (1–5)	2.0 (1.0)	3.0 (1.0)	3.0 (1.0)	2.0 (2.0)	13.85	<0.001
Balance when walking (1–5)	2.0 (2.0)	3.0 (1.0)	3.0 (1.0)	2.0 (2.0)	21.97	<0.001
Balance when sitting (1–5)	1.0 (0)	1.0 (1.0)	1.0 (1.0)	1.0 (1.0)	5.38	0.06
Vision						
Double vision (1–4)	1.0 (1.0)	1.0 (1.0)	1.0 (1.0)	1.0 (1.0)	2.84	0.24
Vision problems—right eye (1–4)	1.0 (1.0)	2.0 (1.0)	2.0 (1.0)	1.0 (1.0)	8.46	0.01 *
Vision problems—left eye (1–4)	1.0 (1.0)	2.0 (2.0)	2.0 (2.0)	2.0 (1.0)	7.21	0.02 *
Blind spot in vision (1–4)	1.0 (0)	1.0 (0)	1.0 (0)	1.0 (0)	2.65	0.26
Face and neck						
Muscle weakness—right side (1–4)	1.0 (0)	1.0 (1.0)	1.0 (1.0)	1.0 (0)	11.18	<0.01
Muscle weakness—left side (1–4)	1.0 (0)	1.0 (1.0)	1.0 (1.0)	1.0 (0)	7.73	0.02 *
Feeling—right side (1–5)	1.0 (0)	1.0 (2.0)	1.0 (2.0)	2.0 (0)	6.9	0.03 *
Feeling—left side (1–5)	1.0 (0)	1.0 (1.0)	1.0 (1.0)	2.0 (0)	3.81	0.14
Ability to speak (1–6)	2.0 (1.0)	1.0 (1.0)	1.0 (1.0)	1.0 (1.0)	3.27	0.19
Ability to swallow liquids and solids (1–4)	1.0 (0)	1.0 (1.0)	1.0 (1.0)	1.0 (0)	2.94	0.22
Hearing problems (1–4)	1.0 (0)	1.0 (1.0)	1.0 (1.0)	1.0 (0)	4.15	0.12
Bowel and bladder						
Urinary urgency (1–5)	2.0 (3.0)	4.0 (1.0)	4.0 (1.0)	3.0 (3.0)	12.69	<0.001
Urine leak (1–5)	1.0 (1.0)	2.0 (1.0)	2.0 (1.0)	1.0 (1.0)	3.03	0.21
Wearing a pad or use a urinal (yes/no) ¶	26/79	9/16	4/17	39/112	χ^2^ = 1.10	0.49
Start to urinate (1–6)	1.0 (1.0)	2.0 (2.0)	2.0 (2.0)	2.0 (1.0)	4.65	0.11
Constipation (1–3)	2.0 (1.0)	2.0 (1.0)	2.0 (1.0)	2.0 (1.0)	1.55	0.45
Bowel frequency (1–4)	1.0 (1.0)	1.0 (1.0)	1.0 (1.0)	1.0 (1.0)	1.38	0.06
Mood and thinking ability						
Cognitive abilities (1–5)	2.0 (1.0)	2.0 (0)	2.0 (0)	2.0 (1.0)	4.73	0.09
Fatigue (1–4)	2.0 (0)	3.0 (1.0)	3.0 (1.0)	2.0 (1.0)	9.37	0.01 **
Little interest or pleasure in doing things (1–4)	2.0 (1.0)	2.0 (2.0)	2.0 (2.0)	2.0 (1.0)	3.71	0.15
Feeling down, depressed, or hopeless (1–4)	2.0 (1.0)	2.0 (2.0)	2.0 (2.0)	2.0 (1.0)	3.54	0.17

Abbreviations: Results for each symptom (except *Wearing a pad or use a urinal*) are presented as median (IQR); *—next to symptom degrees of each scale are listed; ^†^—how much strength do participants have to raise each arm and leg in the air; ^††^—do muscle spasms or stiffness make it hard for participants to use arms and legs; ¶—wearing a pad or using a urinal is a categorical variable, so hi square test (χ^2^) is used. The brackets (1–3; 1–4; 1–5; 1–6) indicate the degree on the Likert scale on which the subject gave answers to each question concerning disability; *—*p* < 0.05; **—*p* < 0.01.

## Data Availability

The data presented in this study are available on request from the corresponding author. The data are not publicly available due to privacy restrictions.

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
