# Peer review of "Symptom-Level Disability Status Assessed with an Electronic Unsupervised Patient-Reported Expanded Disability Status Scale (ePR-EDSS) in Multiple Sclerosis Patients—The Example of Croatia"

_jcm, 2022, doi:10.3390/jcm11144081_

Round 1
Reviewer 1 Report
line 63. Please explain better the instrument of ePR EDSS, in what consist and how it was validated. Describe also other online instruments used to evaluated remotely patients EDSS such as online validated PDDS (Lavorgna L, Miele G, Petruzzo M, Lanzillo R, Bonavita S. Online validation of the Italian version of the patient determined disease steps scale (PDDS) in people with multiple sclerosis. Mult Scler Relat Disord. 2018 Apr;21:108-109. doi: 10.1016/j.msard.2018.02.014. Epub 2018 Feb 12. PMID: 29547779.)
line 78. Why did you select 147 patients? Did you calculate a simple size? Other similar studies have a similar population ?
Discussion: Did you discuss in discuss section about digital therapeutics ? ePR EDSS could be considered as a tool for digital health? which place could be roled by ePR EDSS ? ( Abbadessa G, Brigo F, Clerico M, De Mercanti S, Trojsi F, Tedeschi G, Bonavita S, Lavorgna L. Digital therapeutics in neurology. J Neurol. 2022 Mar;269(3):1209-1224. doi: 10.1007/s00415-021-10608-4. Epub 2021 May 20. PMID: 34018047; PMCID: PMC8136262.)
Author Response
Dear Reviewer 1,
the responses to the comments are attached in a word file.
Kind regards,
Corresponding Author

Reviewer 2 Report
In the present study, the authors applied an ePR-EDSS questionnaire to 147 patients with MS in Croatia. The conclusion of the study is that ePR-EDSS showed more pronounced problems in PPMS and SPMS compared to RRMS concerning overall wellness-functional abilities, mobility, sensation and strength in legs, strength in the right arm, coordination, vision, muscle weakness of both sides of the face, and feeling on the right side of the face, the urgency to urinate and fatigue.
Patient reported outcomes are undoubtedly a valid tool for assessing the disability status and health conditions but as rightly pointed out by the authors themselves, the major limitation of this study is that it does not provide any correlations between the score obtained through the questionnaire and the main clinical tools to assess the disease and does not provide any information about the effectiveness of treatments.
Minor:
Materials and Methods
Results showed in the Participant section (lines 78 and 79) should be moved in Results section.
Results
Table 1
· Results of categorical variables should be presented as frequencies and percentages and not only as percentages
· Variables with a non-gaussian distribution should be presented as median (IQR) and NOT as median±IQR
· Authors should add some details about missing values
Figure 1: I suggest to delete it
Table 2 and Table 3
· Results should be presented as frequencies and percentages and not only as percentages
Table 4
· It could be merged with Table 1
· There is a typo in presenting “MS type not know data”: N should be 17 and not 7
Table 4 and Table 5
· Please provide the correct p-values and not only >0.05
Table 6
· Since the test used is a non-parametric test, I would expect to find the variables summarized as medians and IQRs and not as means and standard deviations

Author Response
Dear Reviewer 2,
the responses to the comments are attached in a word file.
Kind regards,
Corresponding Author
